# Defining Priority Areas for the Sustainable Development of the Desalination Industry in Chile: A GIS Multi-Criteria Analysis Approach

**Enzo García-Bartolomei** [1,2,3,*], **Vanesa Vásquez** [2,3], **Gonzalo Rebolledo** [1,4], **Andrés Vivallo** [3], **Tomás Acuña-Ruz** [3], **José Rebolledo** [2], **Rodrigo Orrego** [5] **and Ricardo O. Barra** [6]

1   Programa de Doctorado en Ciencias Ambientales, Facultad de Ciencias Ambientales y Centro EULA-Chile, Universidad de Concepción, Concepción 4030000, Chile; grebolle@uct.cl
2   IGA Consult, Providencia, Santiago 7500000, Chile; vanesa@igac.cl (V.V.); jrebolledo@igac.cl (J.R.)
3   Bloom Alert, Providencia, Santiago 7500000, Chile; andres.vivallo@bloomalert.com (A.V.); tomas@bloomalert.com (T.A.-R.)
4   Laboratorio de Planificación Territorial, Departamento de Ciencias Ambientales, Facultad de Recursos Naturales, Universidad Católica de Temuco, Temuco 4780000, Chile
5   Aquatic Toxicology Laboratory (AQUATOX), Instituto de Ciencias Naturales Alexander von Humboldt, Universidad de Antofagasta, Antofagasta 1240000, Chile; rodrigo.orrego@uantof.cl
6   Facultad de Ciencias Ambientales y Centro EULA Chile, Centro de Recursos Hídricos para la Agricultura y Minería (CRHIAM), Instituto Milenio SECOS, Universidad de Concepción, Concepción 4030000, Chile; ricbarra@udec.cl
*   Correspondence: enzo@bloomalert.com

**Abstract:** The climate crisis is rapidly provoking water scarcity in several areas of the planet, where an exponential growth of the seawater desalination industry is expected. In this context, multiple efforts are currently under development to reduce potential impacts and promote the sustainability of this industry. The selection of a suitable site for a desalination plant is critical to ensure operational continuity and the environmental sustainability of its processes, optimizing the plant's productive performance and thus safeguarding water security for final users. In this study, we apply a GIS-based Multi-Criteria Analysis (GIS-MCA) approach to explore and assess potential areas suitable for the construction and operation of desalination plants in Chile. Different environmental, social, and technical criteria were evaluated and weighted by expert criteria using the Analytic Hierarchy Process (AHP) methodology. From a total of 114,450 km$^2$ analyzed, only 4.54% of the territory was classified as highly suitable, demonstrating the scarcity of space available to meet the growth expectations for the industry. These results suggest that GIS-based analysis provides a practical solution to determine suitable areas for developing desalination plants, highlighting the need to define priority areas for the sustainable development of the desalination industry in Chile with the required capacity to reach the national water security goals for the following decades.

**Keywords:** AHP; plant location; reverse osmosis; seawater; site suitability

## 1. Introduction

The climate crisis has provoked drastic changes in weather patterns, and the resulting scarcity of water is one of the most complex challenges facing humankind. As an alternative, seawater (SW) desalination and water reuse technologies have become the most feasible solutions to mitigate this issue [1]. According to the International Desalination Association, worldwide, more than 18,000 desalination plants produce around 80 million cubic meters of fresh water daily, and it is expected that global desalination capacity will increase almost two-fold by 2050 [2].

Site selection is crucial when planning for a SW desalination plant, requiring an adequate surface area, topography, and geological features that allow the appropriate

construction and operation of critical infrastructures, such as marine structures and inland facilities [3,4]. In addition, supply access and interconnections to the power grid and fresh-water network should be economically and technically viable. Proximity to highways and roads is also taken into account, although not considered a bottleneck factor. Furthermore, potential environmental, social, and economic impacts are relevant to ensure community acceptance and minimize risks during early stage project development [3]. Thus, defining specific criteria and stakeholders' requirements should be established to guarantee the optimal planification of the SW desalination plant settlement.

Different site-selection approaches are available, where some methods may be more desirable for a particular project to highlight facility-specific considerations or technical judgment from a multidisciplinary team [5]. To date, different site-selection methods have been described, such as TOPSIS [6], Fuzzy [7], Delphi [8], and analytical hierarchy process (AHP) [9]. These models can be strengthened by the use of geographic information systems (GIS) that allow a spatial projection of the territory based on the available public information [10,11]. In 2017, Sepehr, Fatemi, Danehkar, and Moradi [8] proposed 63 locations suitable for installing desalination plants on the coastal areas of the Persian Gulf and Oman Sea, based on an integrated Delphi-GIS model, highlighting that environmentally sensitive areas and SW quality were the most important criterion. Criteria for selecting suitable sites for desalination units construction in South Iran comprised environmental protection area, topography, atmospheric information, oceanography, energy supply, national gas transmission lines, and communication lines, amongst others [12]. Conversely, a multi-criteria analysis (MCA) tool coupled with GIS can be used to create interactive maps highlighting specific areas idoneous for a desalination plant construction [13]. Indeed, the integration of these two approaches has been increasingly applied in order to cope with several environmental and urban assessment problems, where MCA has proved to be a useful decision-support tool by combining diverse factors to produce a practical evaluation index [4,14,15]. For instance, a GIS-MCA method was employed to plan and manage the existing and proposed brackish-water desalination plants in the Gaza Strip [16]. The study revealed that 65% of this area was unsuitable for the construction of desalination plants, and that 53% of the already operating plants showed potential risks to the environment [16]. However, there is still a lack of knowledge of what type of approach and methodology might be the most convenient combination involved in the development and prioritization of factors and/or restrictions to choose appropriate areas to build SW desalination plants.

In Chile, the ongoing megadrought event that has extended between 2010 and 2021 has become an impediment to economic growth for the arid and semi-arid regions [17]. To address this challenge, desalination plants have been proposed as one of the most reliable solutions [18]. According to the Chilean Ministry of Public Works, there are 24 SW desalination plants currently operating and 22 under construction/planning stages mainly in coastal zones in central and northern Chile. North Chile has a vast coastline extending across more than 2000 km. It is characterized by the presence of coastal mountains that rise close to the shoreline with steep cliffs, where only few beaches and natural harbors exist [19]. Moreover, the Atacama desert (>100,000 km$^2$) covers most of the north's territory of Chile and has been rapidly advancing to the central zone of the country at an estimated rate of 0.4 km per year [20]. Therefore, the impending scenario points to the imminent propagation of the desalination industry to alleviate the water crisis situation in Chile.

In this study, we use the analytical and predictive advantages provided by GIS-MCA to assess the multifactorial suitability of potential sites and select priority areas for the construction of desalination plants between the regions of Arica and Parinacota, and the Metropolitan Region of Chile. Multi-criteria spatial evaluation and decision tools, such as hierarchical process analysis (AHP) approaches, are combined as a guide for planning desalination plants settlements. The results suggest that GIS-based analysis provides a cost-efficient solution to determine suitable areas for developing desalination plants, promoting desalination as a feasible solution to the national and global water security challenges.

## 2. Materials and Methods

### 2.1. Study-Site Features: Chilean Far- and Near -North, and Central Zones

The study site comprised the Arica and Parinacota (362,962 E, 7,956,813 S) to the Metropolitan regions (346,816 E, 6,296,840 S) of continental Chile, corresponding to the Chilean Far- and Near -North zones, together with part of the Central zone of the country. This represents a total surface area of 332,825 km$^2$, representing 39.5% of the total national territory, with an estimated associated population of 11,136,000 inhabitants. The Far North is an extremely arid zone containing the Atacama Desert, where most of the population lives in the coastal area. The landscape is characterized by crisscrossing hills and mountains composed of several types of minerals. The coastline is mainly dominated by coastal cliff with elevations over 1000 m. Average annual temperatures oscillate between 14 and 20.5 °C, with almost no precipitation events, except for some areas nearby the Andes mountains [21]. The Near North is a semi-arid region with an annual average precipitation of 25 mm subjected to drought events and mean temperatures between 12 and 18.5 °C. Coastal elevations are often broken by river valleys, and microclimatic conditions appear due to maritime fogs, which can penetrate inland. The Central zone possesses a Mediterranean-type climate with a progressive increment in precipitation events, and annual temperatures between 7.5 and 19.5 °C. This zone is the most densely inhabited part of Chile, which includes the country capital Santiago and Valparaiso region with more than 8 million people. Topography includes the coastal range mountains that form the central valleys and several coastal plains sometimes interrupted by the coastal cliff [22].

### 2.2. Establishment of Selection Criteria

In order to define the variables to be used in the multi-criteria analysis, two types of criteria were considered: factors, that is, criteria that improve the suitability of a specific alternative for the activity under consideration, and constraints that refer to spatial limitations that, for this study, restrict the distribution area of the desalination industry in Chile [23]. Based on this, six factors were established: altitude, distance to the coastline, land slope grade, distance to populated settlements, distance to power supply lines, and distance to the road network. In contrast, four constraints were established: conservation areas with a buffer area of 3000 m, indigenous communities with a buffer area of 1000 m, altitude, and land slope grade [3,24]. The last two have mixed features; that is, under a certain numerical range they are recognized as a factor and in another numerical range they behave as constrains. Thus, altitudes above 1500 m.a.s.l. and slopes above 15° were considered as constrains. Considering these restrictions is expected to minimize the effect of installing desalination plants near wild protected areas involving natural monuments, national parks, and reserves, established by Supreme Decree No. 531/67 of the Chilean law.

### 2.3. Systematization of Geographical Information

Factor definition used for the construction of spatial suitability models was conducted based on available information at the Chilean Environmental Impact Assessment Service (SEIA, https://www.sea.gob.cl/, accessed on 10 May 2021) and the Registry of Desalination Plants and Seawater Pumping Systems 2017/18. Both operating and planned projects were georeferenced in WGS 1984, UTM zone 19S coordinates. Additionally, spatial data were obtained from the Chilean Natural Resources Information Center (https://www.ciren.cl/, accessed on 10 May 2021), the National Energy Commission (https://www.cne.cl, accessed on 10 May 2021), the Chilean National Corporation for Indigenous Development (https://siic.conadi.cl/, accessed on 10 May 2021), the Chilean Meteorological Directorate (http://geonode.meteochile.gob.cl/, accessed on 10 May 2021), Chilean Spatial Data Infrastructure (https://www.ide.cl, accessed on 10 May 2021), and the Chilean Register of Protected Areas (http://areasprotegidas.mma.gob.cl/, accessed on 10 May 2021) (Table 1).

**Table 1.** Spatial data sources.

| Geospatial Data | Source | Year |
|---|---|---|
| Wild Areas Protected by the State | Register of Protected Areas, Chilean Environmental Minister | 2021 |
| Populated areas | Chilean Meteorological Directorate | 2016 |
| Indigenous communities | National Corporation for Indigenous Development | 2017 |
| Administrative Political Division | Geospatial Data Infrastructure | 2019 |
| Coastline | Geospatial Data Infrastructure | 2020 |
| Power supply grid | National Energy Commission | 2016 |
| Alos Palsar Digital Elevation Model | Natural Resources Information Center | 2019 |
| Road network | Geospatial Data Infrastructure | 2019 |

*2.4. Multi-Criteria Factor Weight Analysis Survey*

To assess the preference between factor-type criteria, the methodology of analytical hierarchical process (AHP) was applied [25]. For this purpose, an analysis of hierarchies between variables using a square matrix, which compares the alternatives on a scale of 1 to 5, was performed to define relative priority, both one over another or equal priority [26]. Based on the AHP guidelines, a survey was constructed and sent to 22 international experts in the field of desalination. The survey was made up of three sections and consisted of a total of 22 multiple-choice questions that aimed to answer the following inquiries: what was the relationship between the surveyed and the desalination industry (1 question), which factors were considered relevant or not in the multicriteria analysis (6 questions), and what factors had greater weight when compared to each other (15 questions). To determine the relationship of the surveyed experts with the industry and factor relevance, frequency was used as a description measure. To determine the weight of the factors when comparing them among themselves, the mode was applied as a central trend parameter. In cases where bi-modalities were obtained, the core research group in charge of the surveys decided how to proceed with these specific cases [25].

To measure the opinions and decisions consistency made by the panel of experts, quantitative indexes were used, such as the consistency index (*CI*), the random index (*RI*) and the consistency ratio (*CR*) [25,26]. The degree of consistency obtained depends on whether the information collected through the AHP methodology is reliable or the survey process must be repeated.

The first index to be measured is the consistency index (*CI*), which is understood as the measurement of consistency when comparing between pairs and is expressed as follows:

$$CI = \frac{\gamma_{max} - n}{n - 1}$$

where $\gamma_{max}$ is the largest eigenvalue and *n* is the number of criteria. The random index (*RI*) can be calculated mathematically as the average *CI* of the largest sample from randomly generated comparison matrices [25]. It is expressed through the following expression:

$$RI = \frac{1.98\,(n - 2)}{n}$$

Finally, the consistency ratio is established as the ratio between the consistency index and the random index [25], as follows:

$$CR = \frac{CI}{RI}$$

Consistency ratio assessments greater than 0.10 were considered inconsistent. Conversely, valuations less or equal to 0.10 were considered to have a reasonable level of consistency.

### 2.5. Implementation of the Multi-Criteria Assessment

A proximity analysis was performed using ArcMap 10.3 to determine the linear distance that exist between each desalination project towards the different study factors. Once the distances between the projects and the factors were obtained, these distances were categorized into five groups, which referred to the potential use of the land for the installation of desalination plants (Table 2). Since all vector spatial data presented differences in scale, the layers were rasterized using an Euclidean distance tool and considering the categories mentioned in Table 2 with a spatial resolution of $100 \times 100$ m (equivalent to 1 hectare). Constraints were rasterized to the same spatial resolution as factors ($100 \times 100$ m). However, unlike the factors, constraints have pixel values equal to 0 and 1. The value 0 refers to areas that are not susceptible to the development of the desalination industry, because they present limitations in terms of the presence of conservation areas, indigenous communities, high slopes and altitudes above 1500 m.a.s.l., while value 1 represents territory that could potentially be used for the desalination industry.

**Table 2.** Factor categorization based on potential use.

| Suitability | Pixel Value | Decile |
|---|---|---|
| Very high | 1 | 10 |
| High | 2 | 25 |
| Medium | 3 | 50 |
| Low | 4 | 75 |
| Very low | 5 | >75 |

Multi-criteria evaluation of raster images was combined with the weighted linear combination method (WLC) through a raster overlay operation in GIS [27], assuming that all overlaid layers have a standard scale. The WLC analysis considers the weighted sum of the factors from surveys and the AHP analysis, plus the product from restriction criteria integrated into the mathematical formula as follows:

$$S = \sum_{i=1}^{n} ( F_i W_i) \prod_{k=1}^{n} (C_k)$$

where $S$ is suitability, $F_i$ corresponds to factors, $W_i$ is the normalized weight, and $C_k$ to restrictors. The resulting raster image, covered pixel values from 6 to 30, were re-classified into 5 homogeneous categories, which presented the definitive potential areas for the desalination industry settlements. In order to have a better visualization of the cartographic areas, the speckled effect was eliminated, through the ArcMap tool "focused statistics", to obtain equal neighborhood weighting range to nine pixels.

## 3. Results

### 3.1. Selected Factors and Constraints

The considered social, technical, and environmental variables were chosen on the basis of a literature review and expert judgement [3,23,24]. These criteria were selected to find areas that allow the construction and operation of desalination plants with the lowest technical cost and environmental impact. Thus, the following factors were used to evaluate spatial feasibility models: altitude, distance to the coastline, land slope grade, distance to populated settlements, distance to power supply lines, and distance to the road network (Table 3). Based on expert criteria and observed desalination project's location, optimal distance ranges were configured for each factor, which allowed categorizing the spatial feasibility of a desalination plant being built and operated under those conditions. Subsequently, raster maps were built to visualize the spatial distribution of each selected parameter (Figure 1). On the other hand, constraints were defined based on published research [3,23,24], considering natural conservation areas (buffer area = 3000 m), indigenous communities (buffer area = 1000 m), altitude (above 1500 m.a.s.l.), and land slope (above

15°). Thus, of a total of 332,825 km², encompassing the whole study site, the application of restrictor parameters reduced the area to a total of ~114,450 km² (Figure 2).

**Table 3.** Defined distances between positive factors and desalination project feasibility.

| Feasibility | Factors | | | | | |
|---|---|---|---|---|---|---|
| | Altitude [m.a.s.l.] | Populated Settlements [m] | Coastline [m] | Power Supply Grid [m] | Land Slope [°] | Road Network [m] |
| Very high | ≤300 | >5000 | ≤100 | ≤200 | ≤3 | ≤100 |
| High | 600 | 5000 | 200 | 500 | 5 | 200 |
| Medium | 900 | 2000 | 300 | 1500 | 8 | 400 |
| Low | 1200 | 400 | 800 | 5000 | 12 | 900 |
| Very low | 1500 | ≤100 | >2500 | >5000 | 15 | >900 |

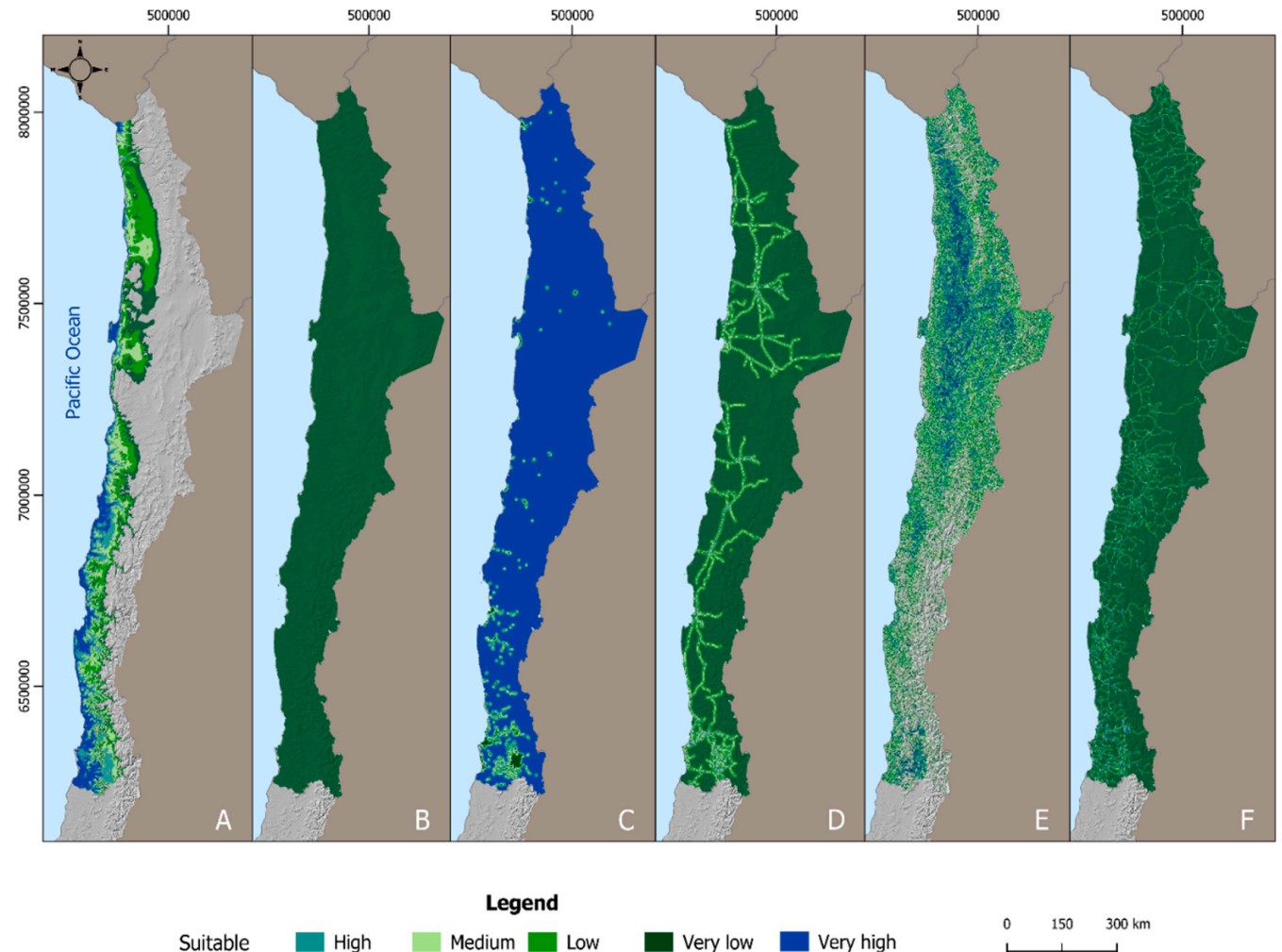

**Figure 1.** Factors' spatial distributions in northern Chile. Altitude (**A**), coastline (**B**), populated settlements (**C**), power supply grid (**D**), land slope (**E**), and road network (**F**). Colors represent suitability for the installation of desalination plants according to the selected criteria.

### 3.2. Survey Application with a Multi-Criteria Approach

A survey to estimate the weights of each factor was conducted of a total of 22 desalination industry experts, including engineering, operation, and maintenance services; academia/research and development professionals; environmental science professionals; and state agencies (Figure 3A). Survey results consisted of ranked criteria based on the significance and severity impacts of each factor, where frequency and the mode were used as measures of dispersion and central tendency, respectively, to determine the relationship

of the respondents with the relative weight of the factors (Figure 3B). In the case of bimodalities (e.g., altitude vs. distance to populated settlements), values were decided through a sensitivity analysis according to the Saaty scale [25], testing values by comparison between pairs and deciding on those generating more reliable consistency indices (w ≤ 0.10). Table 4 shows the preferred factors governing site suitability by the surveyed experts and their normalized weights. Preferred factors correspond to distance to the coastline (w = 0.351), followed by altitude (w = 0.289). These factors will have a greater incidence when identifying high-suitability emplacement areas, compared to lower normalized weight factors, such as populated settlements (w = 0.057) and distance to the road network (w = 0.061) (Table 4). When comparing the responses from surveyed experts according to their related job in the desalination industry, there was no preference bias in the weighted criteria. Furthermore, consistency was tested by employing an AHP matrix, where the consistency index was *CI* = 0.08 and the random index *RI* = 1.24. Hence, consistency ratio equals *CR* = 0.064, reflecting coherence between the analyzed data.

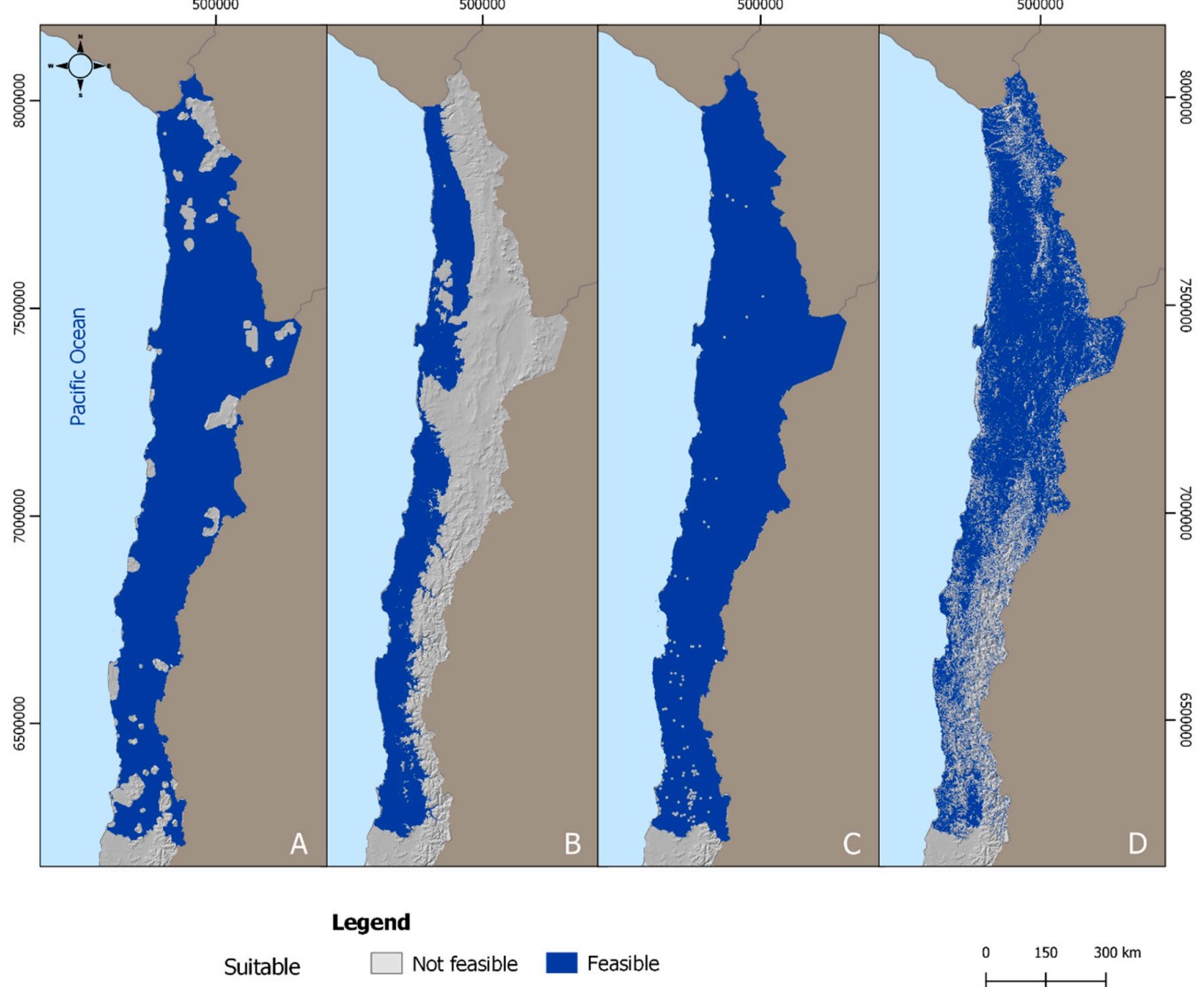

**Figure 2.** Spatial constraints distribution in northern Chile. Conservation areas (**A**), altitude (**B**), indigenous communities (**C**), and land slope (**D**). Blue zones represent potential areas in which desalination plant construction is feasible.

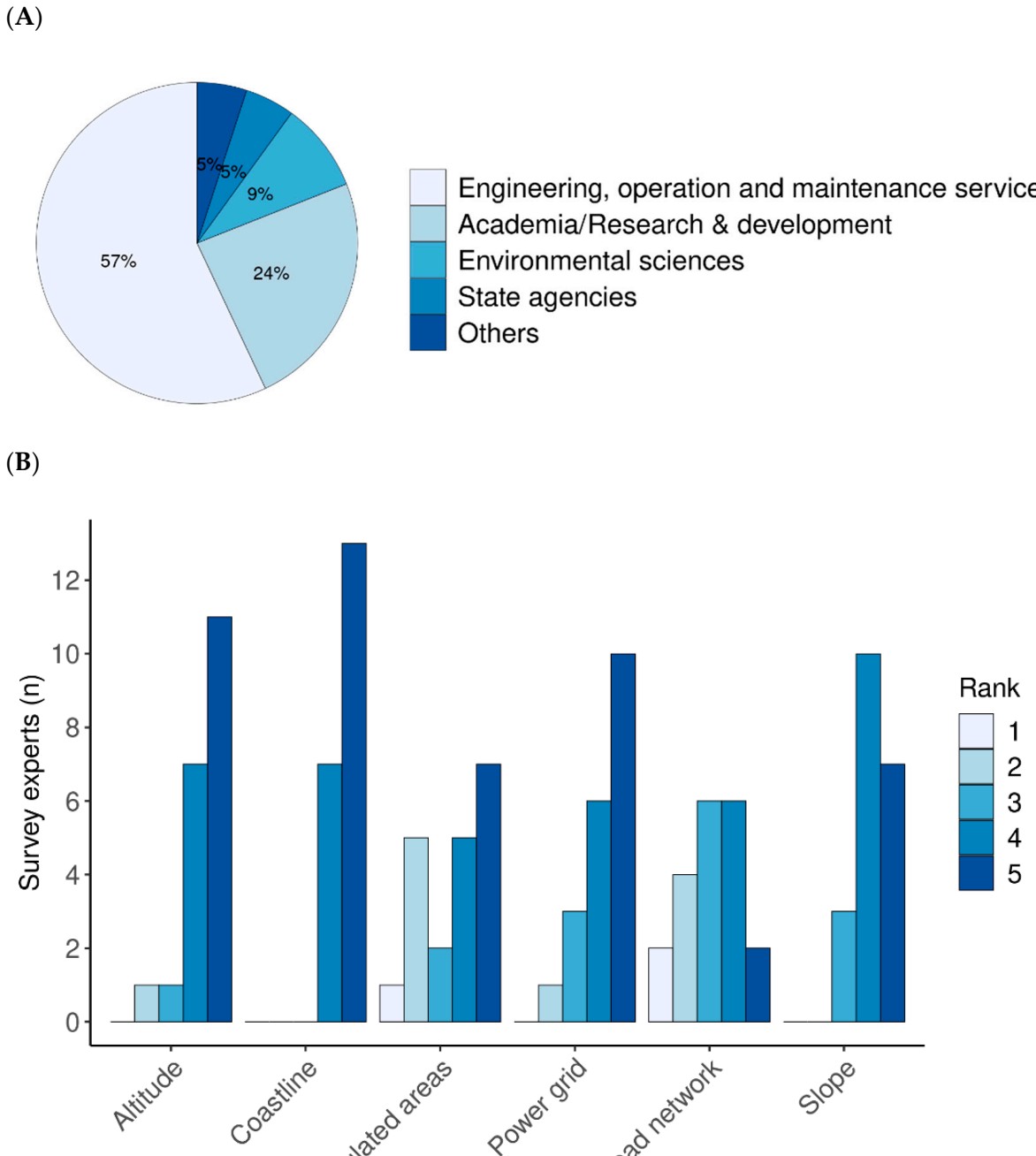

**Figure 3.** Survey features with a multi-criteria approach. (**A**) Pie chart showing the relative number and the field of the different professionals surveyed. (**B**) Assessment of the evaluated factors.

**Table 4.** Factor weight determination through Analytic Hierarchy Process.

| Criteria | Altitude | Populated Areas | Coastline | Power Supply | Road Network | Slope | W |
|---|---|---|---|---|---|---|---|
| Altitude | 1 | 5 | 1 | 2 | 5 | 4 | 0.289 |
| Populated areas | 1/5 | 1 | 1/5 | 1/2 | 1/2 | 1 | 0.057 |
| Coastline | 1 | 5 | 1 | 4 | 5 | 6 | 0.351 |
| Power supply | 1/2 | 2 | $\frac{1}{4}$ | 1 | 5 | 5 | 0.176 |
| Road network | 1/5 | 2 | 1/5 | 1/5 | 1 | 1/2 | 0.061 |
| Slope | 1/4 | 1 | 1/6 | 1/5 | 2 | 1 | 0.065 |

### 3.3. GIS-MCA Site Suitability Model

The normalized weight multi-criteria model shows a clear definition of the areas with the most significant potential versus those with the least susceptibility to installing desalination plants. The distribution map generated shows that potential areas for the industry are present in all the regions under study (Figure 4). As expected, those areas with the most significant installation potential are mainly associated with the distance to the coastline. Therefore, the sectors with the most significant potential for installation are the coastline around the Mejillones Bay in the Antofagasta region (II), between Copiapó and Vallenar in the Atacama region (III), and the southern coastal edge of the Valparaíso region (V) towards the Metropolitan region (Figure 4).

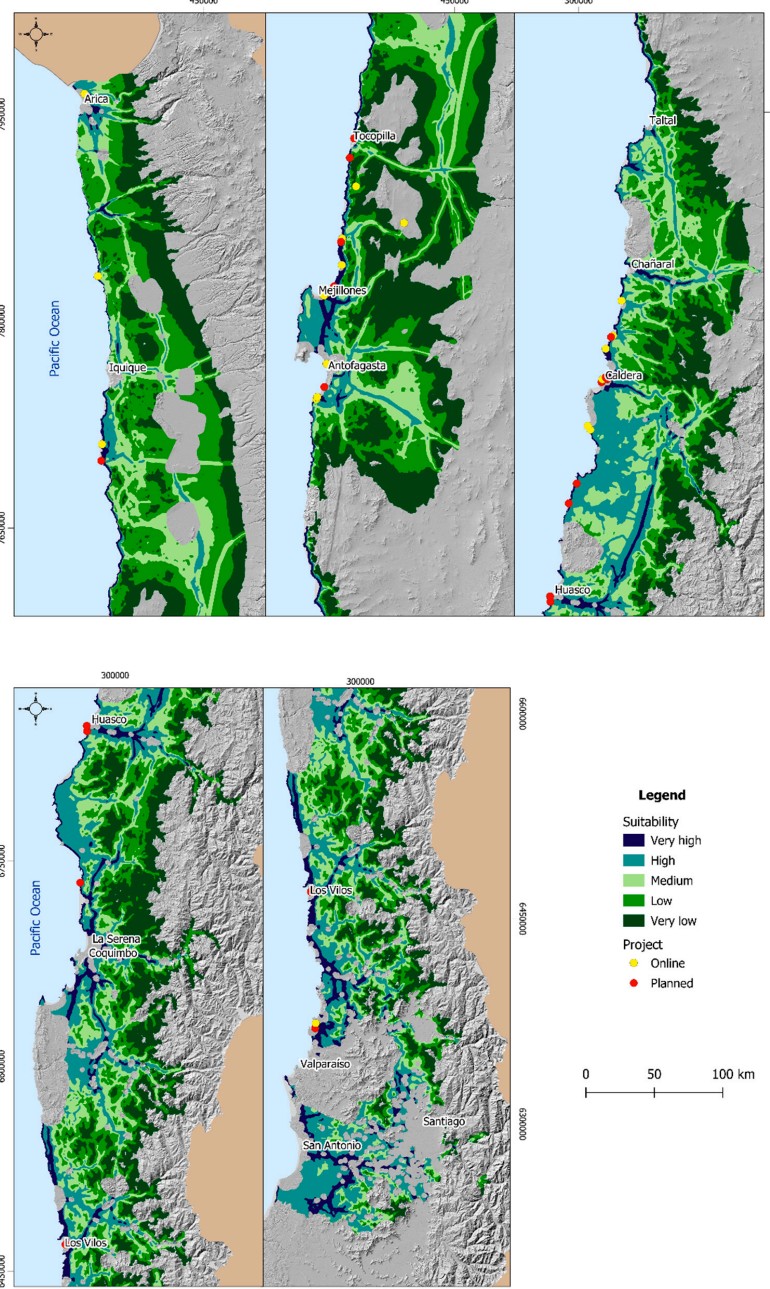

**Figure 4.** GIS-MCA map of northern Chile showing potential areas suitable for the installation of desalination plants, and the current operative or planned desalination plants projects in the same territory.

Potential areas in km² calculated in regard to their suitability for the desalination industry are shown in Table 5. The total area with "very high" and "very low" suitability ranking are 5195 km² (4.54%) and 35,277 km² (30.84%) for the whole northern Chilean territory, respectively. The regions of Coquimbo (IV), Antofagasta (II), and Atacama (III) were highlighted as those with the most suitable areas ("very high" rank) for desalination plant settlements accounting for 1241 km² (1.08%), 1053 km² (0.92%), and 1039 km² (0.91%), respectively.

**Table 5.** Potential areas suitable for desalination plants installation by region.

| Region | N° | Area | | | | | | | | | | | |
|---|---|---|---|---|---|---|---|---|---|---|---|---|---|
| | | Very High | | High | | Medium | | Low | | Very Low | | Total | |
| | | km² | % | km² | % | km² | % | km² | % | km² | % | km² | % |
| Arica y Parinacota | XV | 110 | 0.10 | 523 | 0.46 | 894 | 0.78 | 1357 | 1.19 | 1271 | 1.11 | 4155 | 3.63 |
| Tarapacá | I | 297 | 0.26 | 877 | 0.77 | 3686 | 3.22 | 7782 | 6.80 | 4606 | 4.02 | 17,247 | 15.07 |
| Antofagasta | II | 1053 | 0.92 | 2301 | 2.01 | 5206 | 4.55 | 9325 | 8.15 | 14,485 | 12.66 | 32,368 | 28.27 |
| Atacama | III | 1039 | 0.91 | 6373 | 5.57 | 7524 | 6.57 | 717 | 6.27 | 7081 | 6.19 | 29,187 | 25.50 |
| Coquimbo | IV | 1241 | 1.08 | 3796 | 3.32 | 4377 | 3.82 | 4711 | 4.12 | 5972 | 5.22 | 20,099 | 17.56 |
| Valparaíso | V | 800 | 0.70 | 2138 | 1.87 | 1144 | 1.00 | 992 | 0.87 | 1366 | 1.19 | 6439 | 5.63 |
| Metropolitana | RM | 656 | 0.57 | 2168 | 1.89 | 1012 | 0.88 | 610 | 0.53 | 515 | 0.45 | 4959 | 4.34 |
| **Total** | | **5195** | **4.54** | **18,174** | **15.88** | **23,842** | **20.83** | **31,947** | **27.91** | **35,277** | **30.84** | **114,452** | **100** |

Figure 4 shows the location of 40 desalination plants, both in operation (online) and in the planning stage (planned), of which 21 plants (52.5%) are located in areas of very high suitability, one plant (2.5%) in a high suitability zone and one plant (2.5%) in a medium suitability zone. The remaining 17 plants (42.5%) were located in areas without classification (or not suitable), of which 16 operate within the urban radius of cities and one near a mining site above 1500 m.a.s.l.

The GIS-MCA model can also be used at higher resolution scales, such as bay or communal levels. For example, Figure 5A shows the Mejillones bay in the Antofagasta region (II), one of the hotspots of the desalination industry in Chile. The spatial model presented is consistent with the layout of the desalination plants both online and projected, these being located significantly in areas of very high suitability. At the regional level, the suitability model shows us the spatial planning challenges that interregional projects may encounter, such as what happens with the supply of desalinated water for the metropolitan region, which must cross the Valparaiso region to reach the coastline (Figure 5B).

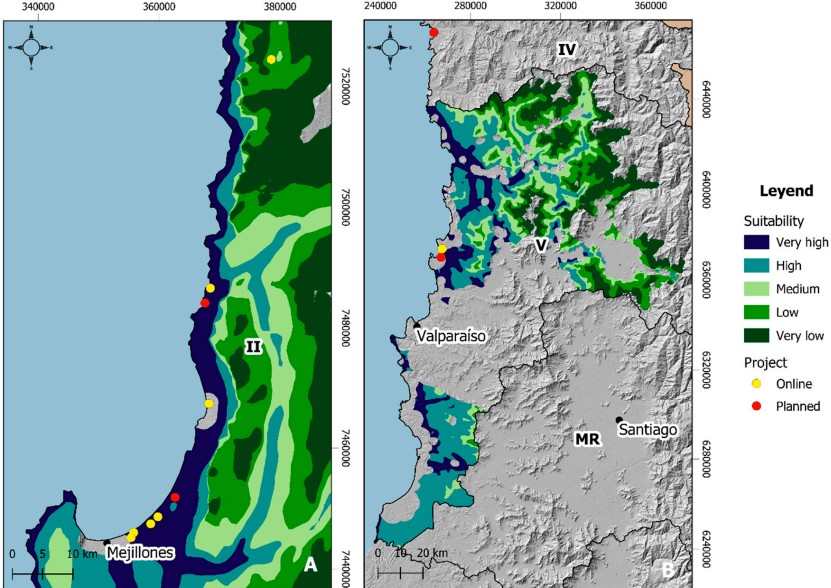

**Figure 5.** (**A**) Suitability map for the Mejillones bay, a hotspot for the desalination industry in northern Chile (II = Antofagasta region). (**B**) Suitability map for the desalination industry in the Valparaiso region (IV = Coquimbo region; V = Valparaíso region; MR = Metropolitana region).

## 4. Discussion

Site selection is a fundamental process for the construction and operation of a desalination plant, ensuring its environmental and operational sustainability [3]. The process of selecting a site for a desalination plant usually involves a complex analysis, including a complex array of features involving socio-economic, technical, and environmental criteria, which must meet strict standards to secure technical and economic viability, minimizing risks in early stage investment and project development [15,23]. The present study successfully implemented an AHP GIS model as a multi-criteria decision tool to evaluate and define the territorial capacity of the north-central region of the country to sustain the development of desalination projects.

Within the different selected factors, the weighted expert criterion using the AHP method indicates that the most essential criteria is proximity to the coastline, which is fundamental to avoiding the increase in costs from raw seawater supply transport, as well as a better brine final disposition directly to the marine environment employing subtidal diffusers with low environmental impact [12]. Secondly, the most crucial valued factor was site altitude. Operational costs and energy consumption associated with pumping water, either for processing or consumption for end-users, are critical for a plant operating expenses (OPEX), so minimizing distribution in height is essential for the project's economic feasibility [28].

Local communities play an essential role in the feasibility of early stage desalination projects, where public perception is a fundamental factor for the operational sustainability and water security of the population depending on this water supply. In a perception study conducted on the California coast, evidence showed that the main factors impacting the negative perception of this industry were potential environmental impacts [29]. Therefore, our model was executed based on protecting all environmentally sensitive and preservation areas present in the study area, considering spatial exclusion buffers that ensure that these areas are not considered potential industrial development zones. Projects have to be extremely careful with the environment in order to not only ensure the operational continuity of its processes, but also to take care of the delicate balance regarding public opinion [30].

From a total of 114,452 km$^2$ analyzed, the GIS-MCA results show the critical situation regarding the availability of space for the development of the industry, where only 4.54% of the territory, equivalent to 5195 km$^2$, was classified as highly suitable and tended to be distributed mainly along the coastline. On the contrary, more than 60% of the territory under study has a low or very low suitability classification, distributed through the east of the country with steep slopes due to the Andes mountains range that dominates the landscape. These results are consistent with results from a recent study in the Gaza Strip, where the MCA models showed that only 2% of the territory was highly suitable for developing desalination plants, compared to 57% of the whole Gaza strip where conditions were unsuitable [16]. This situation is aggravated from a regional perspective, where no region significantly exceeds 1% territorial availability with a very high suitability classification, which is relevant considering that government's water security strategy is based on demand and growth models on a regional scale.

The presence of high-suitability areas distributed in the central valley of the country, even though they are considerably distant from the coast, are valuable for the desalination industry considering that metal mining and agriculture are intensively developed in this region, with both activities driving in water consumption and also being essential for the country's economy. Furthermore, the severe water crisis that Chile is going through is forcing these industries to explore new water sources, such as seawater desalination. For the last two decades, the Chilean copper mining industry has been developing desalination projects, representing more than 70% of the installed desalination capacity in the country, and by the year 2030, it is estimated that seawater will represent 93% of the water used in its production processes [31,32].

The ongoing southwards expansion of the Atacama Desert is exerting intense water pressure on agricultural production in the central zone of the country, where export crops essential for national food security and economy are developed [33]. As a result, agricultural activity, whose irrigation demands are responsible for 75% of water consumption in Chile, is being forced to assess new water sources, such as seawater desalination. Due to the higher costs that this entails, the sustainable supply of crops with desalinated water will face substantial challenges, not only in terms of implementing more complex technical irrigation systems, but also in finding models for the exploitation of desalinated water with multipurpose plants, which will make it possible to face the costs of producing desalinated water among a diverse network of end-users [34].

The mega-drought that the country is experiencing intensely affects the central zone, where the regions of Valparaiso and Metropolitana (main administrative centers of Chile) are the most affected [35]. The capital Santiago de Chile (RM) is the country's leading urban center with 6.2 million inhabitants and faces this year, for the first time in its history, a water rationing program due to the decrease in its water reservoirs [36]. Seawater desalination is a possible solution to the water crisis; however, as it is a Mediterranean region, it must cross the Valparaiso region to access seawater. Our results reveal significant challenges in conducting this type of infrastructure operation, as a high percentage of the Valparaiso region is represented by ecological preservation zones and coastal settlements, with considerable difficulties to build desalination plants and their distribution aqueducts, this without considering that the region has its own water security challenges.

## 5. Conclusions

This study is the first to address the need to identify the space available for the growth of the desalination industry, driven by the demand projections imposed by the current water crisis in Chile. The methodological approach used at a national, regional, or bay resolution level proved to be effective in providing a reasonable logical framework to support decision-making processes regarding the identification of appropriate sites for the installation of desalination plants. From a series of AHP-paired comparison matrices, it was possible to obtain a standardization of criterion scores and various multi-criteria suitability maps capable of integrating environmental, socio-economic, and technical variables, as a tool for the territorial feasibility analysis of the desalination industry in areas of the country with the greatest hydric stress.

Future studies should consider new socio-environmental variables, including coastal water quality, frequency of red tide events, tsunami risk areas, and economic models. Their inclusion will allow the elaboration of multi-criteria restriction maps at a higher spatial resolution, capable of describing in more detail the diversity of environments and territorial challenges faced by the desalination industry in Chile. This study does not replace the need to conduct baseline in situ evaluations to assess potential environmental impacts or the presence of human settlements that may have been left out of the analysis at the scale presented.

Chile has ambitious growth projections for the desalination industry, on which it is basing a large part of its water security strategy. However, this is not consistent with the results obtained in this study, showing a low availability of space on the coast for the development of desalination plants capable of meeting this demand. This situation can lead to increased competition processes for the use of the coastline and, eventually, the construction of plants one near the other could potentially exacerbate the processes of environmental degradation or negative social perception of an industry that today plays a key role in the water sustainability of an entire country.

The results of this study highlight the need to evaluate spatial models for the definition of priority areas for the desalination industry in Chile, which allow safeguarding sufficient space on the coastline for the operation of plants with the desalination capacity required to reach the national water security goals for the next decades.

**Author Contributions:** Conceptualization, E.G.-B.; methodology, E.G.-B., G.R. and V.V.; formal analysis, E.G.-B., V.V., A.V. and G.R.; investigation, E.G.-B. and V.V.; data curation, V.V., A.V., G.R. and T.A.-R.; writing—original draft preparation, E.G.-B.; writing—review and editing, all authors; visualization, T.A.-R., A.V. and V.V.; supervision, R.O.B. and R.O.; funding acquisition, E.G.-B., J.R. and R.O.B. All authors have read and agreed to the published version of the manuscript.

**Funding:** This research was funded by the National Doctorate Scholarship 2017, Grant N° 21171486 from the National Agency for Research and Development of the Chilean Government (ANID), the IDA Channabasappa Memorial Scholarship 2019/2020 and theWater Research Center for Agriculture and Mining (CRHIAM)—ANID/FONDAP/15130015.

**Institutional Review Board Statement:** The participation of an Institutional Review Board approval was not required for this research. The opinion poll developed during this study was voluntary and anonymous, with no personal information being collected, apart from participants' names, publically available contact details, and a record of consent. Therefore, the data is not considered sensitive or confidential; there is no risk of possible disclosures or reporting obligations for the participants, and the information was treated under Chilean Organic Law 19628 on the Protection of Personal Data. Vulnerable or dependent groups were not included, and the research objectives were not likely to upset or disturb participants.

**Informed Consent Statement:** Informed consent was obtained from all subjects involved in the study.

**Data Availability Statement:** Not applicable.

**Acknowledgments:** E.G.-B. would like to thank Fernanda Rodríguez-Rojas, Loreto Zapata and Chris Harrod for their valuable contributions and comments to the draft, and the Chilean Association of Desalination (ACADES) for its continuous support to the development of applied research in the field of desalination. R.O.B. thanks the support of CRHIAM Center, ANID/FONDAP/15130015 and Instituto Milenio SECOS, Millennium Science Initiative ICN_2019_015.

**Conflicts of Interest:** The authors declare no conflict of interest.

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
