# Peer review of "Defining Priority Areas for the Sustainable Development of the Desalination Industry in Chile: A GIS Multi-Criteria Analysis Approach"

_sustainability, doi:10.3390/su14137772_

Round 1
Reviewer 1 Report
The Authors carefully prepared the manuscript, it requires only minor languages corrections. They should also explain how were the areas protected by the buffer zones selected, how was the extent of these areas determined (row 231-233)?
Author Response
Response to Reviewer 1 Comments
Point 1: The Authors carefully prepared the manuscript, it requires only minor languages corrections.
Response 1: Agree. We have, accordingly, revised the whole manuscript with a native English speaker to emphasize this point, and have incorporated language corrections throughout the manuscript.
Point 2: They should also explain how were the areas protected by the buffer zones selected, how was the extent of these areas determined (row 231-233)?
Response 2: Buffer areas were defined based on the following publications
- Tsiourtis, N.X. Criteria and procedure for selecting a site for a desalination plant. Desalination 2008, 221, 114-125
- Shahabi, M.P.; Anda, M.; Ho, G. Influence of site-specific parameters on environmental impacts of desalination. Desalin. Water Treat. 2015, 55, 2357-2363
- Rikalovic, A.; Cosic, I.; Lazarevic, D. GIS Based Multi-criteria Analysis for Industrial Site Selection. Procedia Engineering 2014, 69, 1054-1063
Even though these works were cited in the Materials and Methods section, it is correct to point out that they should be considered in the suggested paragraph by Reviewer 1, so they were added to the mentioned section.
Reviewer 2 Report
In this paper, the authors applied a GIS-based Multi-Criteria Analysis (MCA) approach to explore the best potential areas suitable for the construction and operation of desalination plants in Chile. Different environmental, social, and technical criteria were evaluated and weighted by experts using the Analytic Hierarchy Process method. The manuscript is clear and presented in a well-structured manner.
Is a very interesting and important application, but I believe it is necessary to point out the relevance of this research to the knowledge. Why is the manuscript relevant? What is the contribution of the research beyond the case study (Chile)?
The cited references are recent publications, hovewer, in my opinion, there is a need for complementation. The recent paper by Gholamalifard et al. (2022) (https://doi.org/10.3390/w14101669) and the article by Aydin and Sarptas (2020) (https://doi.org/10.1007/s10098-019-01783-0) can help.
As questionnaires were applied in the research, it would be advisable to present the approval of the ethics committee.
Include the equation reference (line 212).
Author Response
Response to Reviewer 2 Comments
Point 1: Is a very interesting and important application, but I believe it is necessary to point out the relevance of this research to the knowledge. Why is the manuscript relevant? What is the contribution of the research beyond the case study (Chile)?
Response 1: Seawater desalination is one of the current leading solutions to the international water crisis driven by global climate change. However, the feasibility of developing a desalination project does not depend solely on the water demand but a complex matrix of environmental, social, technical, and economic factors. In this sense, the application of GIS tools allows the development of spatially explicit models capable of synthesizing complex multi-criteria decisions and taking into account the opinion of a diverse panel of experts. Beyond the Chilean case study, this research suggests that GIS-based analysis provides a cost-efficient solution to determine suitable areas for developing desalination plants, promoting desalination as a feasible solution to the global water security challenges.
Point 2: The cited references are recent publications, hovewer, in my opinion, there is a need for complementation. The recent paper by Gholamalifard et al. (2022) (https://doi.org/10.3390/w14101669) and the article by Aydin and Sarptas (2020) (https://doi.org/10.1007/s10098-019-01783-0) can help.
Response 2: Citations suggested by reviewer 2 were added to both the introduction and discussion of the manuscript.
Point 3: As questionnaires were applied in the research, it would be advisable to present the approval of the ethics committee.
Response 3: The opinion poll developed during this study was completely voluntary and anonymous, with no personal information being collected, apart from participants´ names, publically available contact details, and a record of consent. The data is not considered to be sensitive or confidential in nature and, hence, there is no risk of possible disclosures or reporting obligations for the participants. Also, vulnerable or dependent groups are not included and the issues being researched are not likely to upset or disturb participants. Before their participation, every participant was informed with the following record of consent:
"Your authorization is requested to participate in the research project entitled: "Defining Priority Areas for the Sustainable Development of the Desalination Industry in Chile: a GIS Multi-Criteria Analysis Approach” whose objective is to develop a strategic decision-making model of multicriteria spatial analysis with GIS, to support desalination plant site suitability analysis. The results of this study will highlight the need to evaluate spatial models for the definition of priority areas for the desalination industry in Chile, allowing to safeguard sufficient space on the coastline for the operation of plants with the desalination capacity required to reach the national water security goals for the next decades. The study will be carried out from June 2021 until May 2022. Participation in this study is completely voluntary, if you do not want to participate in the study, there will be no negative consequences for you. You can withdraw from the study at any time without any consequences. In this opinion poll, no personal or sensitive data will be collected. The answer is completely anonymous, so no data will be available that can identify you, in any case, the information will be treated in accordance with Chilean Organic Law 19628 on the Protection of Personal Data, of August 1999, and its modification by Organic Law 21214. If you have any questions about this research project, you can contact the research team at any moment. If you answer the questions that are proposed to you, it is tacitly understood that you have understood the objective of this study, that you have been able to ask and clarify any doubts that may have initially arisen, and that you agree to participate in the study."
The study was conducted in accordance with the Declaration of Helsinki and in any case, the information was treated in accordance with Chilean Organic Law 19628 on the Protection of Personal Data, of August 1999, and its modification by Organic Law 21214. Considering the above, the participation of an Ethics Committee or Institutional Review Board approval was not required for this manuscript.
Point 4: Include the equation reference (line 212).
Response 4: Modifications were made to the text and formula of the weighted linear combination method according to the reviewer's observations (lines 206-215).